# Benfotiamine protects MPTP-induced Parkinson's disease mouse model via activating Nrf2 signaling pathway

**Kai Wang[1,2], Chao Han[3], Jinwei Yang[4], Wenhao Xu[1], Lei Wang[1], Huaiyu Li[3]\*, Yu Wang[1]\***

**1** Department of Neurology, The First Affiliated Hospital of Anhui Medical University, Shushan District, Hefei, Anhui Province, People's Republic of China, **2** Department of Neurology, The Third Affiliated Hospital of Anhui Medical University, Luyang District, Hefei, Anhui Province, People's Republic of China, **3** Division of Life Sciences and Medicine, Department of Neurology, The First Affiliated Hospital of USTC, University of Science and Technology of China, Luyang District, Hefei, Anhui Province, People's Republic of China, **4** Department of Critical Care Medicine, The Affiliated Fuyang People's Hospital of Anhui Medical University, Chengnanxin District, Fuyang, Anhui Province, People's Republic of China

\* yw4d@hotmail.com (YW); lihy519@126.com (HL)

**Data Availability Statement:** All data from this study are included within the article.

## Abstract

The pursuit of drugs and methods to safeguard dopaminergic neurons holds paramount importance in Parkinson's disease (PD) research. Benfotiamine (BFT) has demonstrated neuroprotective properties, yet its precise mechanisms in PD remain elusive. This study investigated BFT's potential protective effects against dopamine neuron damage in a PD animal model and the underlying mechanisms. The PD mouse model was induced by 5 consecutive MPTP injections, followed by BFT intervention for 28 days. Motor deficits were assessed via pole test, hang test, gait analysis, and open field test, while dopaminergic neuron damage was evaluated through Immunofluorescence, Nissl staining, and Western blot analysis of Tyrosine Hydroxylase (TH) in the substantia nigra and striatum. High Performance Liquid Chromatography quantified dopamine (DA) levels and its metabolites. Genetic pathways were explored using RNA-seq and bioinformatics analysis on substantia nigra tissues, confirmed by qPCR. Activation of the Nrf2 pathway was examined through nuclear translocation and expression of downstream antioxidant enzymes HO-1, GCLM, and NQO1 at mRNA and protein levels. Additionally, measurements of MDA content, GSH activity, and SOD activity were taken in the substantia nigra and striatum. BFT administration improved motor function and protected against dopaminergic neuron degeneration in MPTP mice, with partial recovery in TH expression and DA levels. RNA-seq analysis revealed distinct effects of BFT and the NLRP3 inhibitor MCC950 on Parkinson-related pathways and genes. Control of Nrf2 proved crucial for BFT, as it facilitated Nrf2 movement to the nucleus, upregulating antioxidant genes and enzymes while mitigating oxidative damage. This study elucidates BFT's neuroprotective effects in a PD mouse model via Nrf2-mediated antioxidant mechanisms and gene expression modulation, underscoring its potential as a therapeutic agent for PD.

**Funding:** This study was supported by the National Natural Science Foundation of China (82071460, to YW), and Science and Technology Major Projects in Anhui Province (202103a07020015, to HYL). This research is also funded the grant 2008085QH366 from Natural Science Foundation of Anhui Province (to CH), the grant 2021M690147 (to CH) from the fellowship of China Postdoctoral Science Foundation.

**Competing interests:** The authors have no financial and non-financial competing interests related to this work.

# Introduction

Parkinson's disease (PD) is a prevalent neurodegenerative disorder, ranking second in prevalence only to Alzheimer's disease. It is distinguished by the gradual deterioration of the motor system [1]. The disease is caused by the ongoing deterioration of dopaminergic neurons in the nigrostriatal region and a reduction in DA levels in the brain [2, 3]. While the utilization of medications like levodopa might provide temporary relief for symptoms, it is important to note that the efficacy of these therapies gradually declines over time and is often accompanied by a significant incidence of adverse effects [4]. Consequently, it is crucial to investigate innovative medications that can safeguard neurons and decelerate the advancement of Parkinson's illness to treat it.

Research indicates that there is a certain relationship between dopamine and thiamine. The cerebrospinal fluid levels of total thiamine in PD patients receiving levodopa treatment are significantly higher than those in patients not receiving this medication [5]. Furthermore, thiamine and its derivatives play a crucial role in dopamine synthesis and release, whereas thiamine pyrophosphate can stimulate alpha-ketoglutarate dehydrogenase to promote energy metabolism and induce dopamine release [6]. Studies have also observed lower levels of free thiamine in the cerebrospinal fluid of PD patients compared to normal individuals. Additionally, in the frontal cortex of patients with Parkinson's disease-dementia syndrome, there is a noticeable reduction in thiamine pyrophosphatase activity [5, 7]. Thiamine deficiency may lead to decreased alpha-ketoglutarate dehydrogenase activity, impaired oxidative metabolism, increased oxidative stress, and apoptosis of specific neurons in certain brain regions. It is suggested that thiamine deficiency could potentially be involved in the onset and development of PD [8, 9].

Within this particular setting, the significance of thiamine, sometimes known as vitamin B1, is emphasized. Thiamine plays a crucial role in various important brain processes. Research has shown that individuals with PD typically have low amounts of thiamine in their blood. This suggests that boosting thiamine levels in individuals with PD could potentially have positive effects on PD [5]. Benfotiamine (BFT), a fat-soluble form of thiamine, has garnered significant interest in the treatment and study of neurological disorders because of its exceptional bioavailability and capacity to traverse the blood-brain barrier [6]. Extensive attention has been devoted to research [10–14]. However, the exact mechanisms by which BFT exerts its protective effects in PD mouse models are not yet fully understood [5, 7, 15].

Mouse models produced by MPTP, a toxin that particularly harms nigrostriatal dopaminergic neurons, have become a valuable tool in investigations focused on PD. These models accurately replicate the clinical and pathological characteristics of PD [16]. Oxidative stress plays a crucial role in the development of PD, and it has been proposed that activating Nrf2, a crucial transcription factor that controls cellular resistance to oxidative stress, could be a possible approach to treat PD [17, 18].

The current study employed the MPTP-induced subacute PD model in C57BL/6J mice and implemented BFT intervention therapy, taking into account the aforementioned factors. We conducted a comprehensive investigation to determine if BFT could have neuroprotective benefits on mice with a PD model by activating the Nrf2 antioxidant pathway using RNA sequencing and other techniques. This not only enhances our comprehension of the mechanism by which BFT contributes to neuroprotection, but also introduces novel concepts and avenues for investigating pharmacological therapy for Parkinson's disease.

# Methods

## Mice

Jiangsu Huachuang Xinnuo Pharmaceutical Technology Co., Ltd. donated C57BL/6J male mice that were 8 weeks old and weighed 20–23 g. The mice passed SPF-grade (specific

pathogen free) testing and were given Animal Quality Certificate No. 202266273. They were kept exclusively in an animal facility with controlled temperature and lighting (21–23 degrees Celsius, 12-hour light/dark cycle) and free access to food and water. The animals were habituated to the laboratory for one week before the tests began. All animal research followed national rules for laboratory animal management as well as guidelines established by Anhui Medical University's Laboratory Animal Center. The animal procedures of this study were approved by Biomedical Ethics Committee of Anhui Medical University (Protocol Number: 20200470). All methods were carried out in accordance with relevant guidelines and regulations. This study was carried out in compliance with the ARRIVE guidelines.

## Agent treatments

Following one week of acclimation, groups of mice received intraperitoneal injections of MPTP at a dosage of 30 mg/kg for five consecutive days to induce the PD model. The control group was given the same amount of saline solution containing 0.9% sodium chloride. The mice were given BFT orally at doses of 200 mg/kg and 250 mg/kg for 28 days [19], whereas another group got MCC950 at a dose of 10 mg/kg intraperitoneally for the same duration. All administrations were carried out concurrently (S1 Fig).

## Behavioral tests

To determine the positive impact of BFT on behavioral impairments in the MPTP-induced PD mouse model, mice underwent evaluations using the pole test, hang test, step distance experiment, and open field test. Motor function of all animals was assessed during the 10:00.

## Pole test

The experimental pole-climbing system consisted of a 55cm-long, 1-cm-diameter wooden pole with a 2-cm-diameter wooden ball attached at the top for mouse placement, along with a foam base. The item was packaged in a cushioned cardboard box to avoid any damage. The gauze was wrapped over the pole's surface to enhance friction and prevent slippage. The behavioral trial began two hours after the medication was administered. The duration from when the mouse began moving to when its head completed the turn downward (T-Turn) and the overall time from when the mouse started standing on the ball to when it reached the bottom of the pole (T-Total) was noted. Each mouse performed three pole-climbing trials at a 1-minute interval, and the average time taken was recorded as the experimental score [20, 21].

## Hang test

The experimental setup consisted of a 50 x 50 cm box with padding at the bottom and a 30 cm-tall metal wire with a 2 mm diameter placed horizontally. Mice were placed in the center of a metal wire and instructed to grasp it with both front paws as part of the experiment. The timer started when the mice dropped. Each mouse underwent the test three times, with a one-minute interval between each trial. The mean of the three occasions was subsequently calculated. The mice were trained daily for three days before the actual experiment to reduce errors. The behavioral experiment commenced two hours after administering the medicines on the trial day as scheduled [22].

## Step distance experiment

Arrange a plastic channel measuring 10 x 10 x 50 cm, with closed ends, and place a white paper measuring 10 x 10 x 50 cm at the bottom. Two-thirds of the terminal side of the canal

was enveloped with an opaque cloth. The ink was uniformly distributed onto the hind paws of both mice. The substance was delicately placed at the beginning of the runway and awaited crossing the water. Record mouse footprints upon paper examination. The distance between two successive footprint points on one hind paw on one side from the runway start should be measured after all mice have crawled for a minimum of three steps. The average step length was obtained by adding and averaging three-step distances [23].

## Open field test

To conduct open field experiments, select a chamber of 50 cm by 50 cm by 20 cm with 25 tiny squares on the bottom. The animals needed to be acclimated in the lab for a minimum of three hours before the commencement of the experiment to reduce their fear of the unfamiliar environment. The tests were conducted in a calm environment with uniform and subdued lighting. VisuTrack Animal Behavior Analysis Software was utilized for 5 minutes to track the mice's line crossings and total crawling distance. The software was utilized to assess the mobility of the mice. After testing each mouse, ethanol was used to thoroughly clean the bottom, placements, and sidewalls of the open-field reaction box. After drying, the mice were reintroduced to conduct the tests once more [24].

## Tissue preparations

Mice were anesthetized with sodium pentobarbital (40 mg/kg, i.p) and perfused transcardially with 25 ml of saline. Midbrain substantia nigra and striatum tissues were collected on ice and stored at -80°C for western blotting, real-time fluorescence quantitative PCR (qRT-PCR), RNA sequencing (RNA-seq), Biochemical indicators of oxidative stress(SOD, MDA and GSH) and high-performance liquid chromatography (HPLC). Brains used for NISSL staining and immunofluorescence (IF) staining were first soaked in saline, then perfused with 20 ml of 4% paraformaldehyde, placed into pre-cooled EP tubes of known weight, weighed again, tissue weight was calculated and labeled, and then quickly placed into liquid nitrogen for quick freezing, and then on the next day, brain tissues were taken out to be fixed with 4% paraformaldehyde for more than 24 h. The areas containing the striatum and substantia nigra were then trimmed with a scalpel. areas containing the striatum and substantia nigra were then trimmed with a scalpel and placed into a correspondingly labeled paraffin-embedded box, followed by coronal sectioning (4 um, Leica, Germany). Sections including substantia nigra and midbrain striatum were mounted on slides coated with polylysine and stored at -80°C.

## Nissl staining

A suitable quantity of Nysted stain (Servicebio, GP1043) was incrementally applied to the sections to induce a blue-violet hue in the neurons. The specimens were subjected to a temperature of 65°C for a duration of 10 minutes. Following this, the slices were subjected to a drying process to achieve transparency. They were then sealed using neutral glue and subsequently analyzed using a light microscope (Leica DMI4000 B system) to evaluate the neuronal degeneration and pathological characteristics.

## Immunofluorescence staining

The portions underwent permeation using 0.3% Triton X-100 (Servicebio, China) for a duration of 40 minutes at room temperature. The sections were subjected to three washes with PBS, followed by blocking with 5%Albumin Bovine V (Biosharp, China) at room temperature for 1 hour. Subsequently, the sections were incubated with the primary antibodies, rabbit anti-

TH (1:200, Abcam-Ab137869, USA), at a temperature of 4˚C overnight. The sections underwent a washing step using five PBS, followed by incubation with secondary antibodies at room temperature in the absence of light for a duration of 2 hours. The secondary antibody used was goat anti-rabbit Alexa Fluor 594 (1:400, ZSGB-BIO, China). The sections underwent five washes with PBS, followed by staining with DAPI (Servicebio, China) for a duration of 5 minutes at room temperature. Subsequently, the DAPI was washed out and the sections were covered with antifade mounting media (Servicebio, China). The images were obtained under identical exposure conditions using a fluorescence microscope manufactured by Leica in Germany. The quantification of cells exhibiting fluorescence and the determination of fluorescence intensity were conducted using Image J (version 1.53t, USA).

## Western blotting

The substantia nigra and striatum tissues were homogenized in lysis buffer(150 mM NaCl,50 mM Tris-HCI 264 (pH 7.4),1 mM EDTA,1 mM PMSF,1%sodium deoxy-265 cholate,1%Triton X-100, and 0.1% SDS)and centrifuged at 4˚C 12,000 rpm for 10 min, then boiled for 15 min. Total protein concentration was quantified by Bicinchoninic Acid Assay (Beyotime, China). The isolation of nuclear and cytoplasmic proteins was performed using the Nuclear and Cytoplasmic Protein Extraction Kit obtained from Beyotime Biotechnology (Shanghai, China), according to the manufacturer's protocol. Samples (30ug) were loaded to electrophorese in 10%SDS-polyacrylamide gel at 55 V for 60 min and then at 110V for 60 min. Proteins were transferred onto a PVDF membrane (Immobilon-P, Millipore, USA)at 20 mA for 225 min, and then blocked by 5%non-fat milk for 1.5 h at room temperature (RT). After 15 min wash in tri-distilled water, the membranes were incubated at 4 Covernight with the primary antibodies: rabbit anti-TH(1:5000, Abcam, USA), Nrf2(1:5000, Proteintech, China), HO-1(1:5000, Proteintech, China), GCLM(1:5000, Proteintech, China), NQO1(1:5000, Proteintech, China)and mouse anti-β-actin (1:5000, Proteintech, China). After four PBST washes, the membranes were incubated with the corresponding secondary antibodies: goat anti-rabbit IgG (1:10,000, ZSGB-BIO, China)and goat anti-mouse IgG (1:10,000, ZSGB-BIO, China)for 2 h at RT. The membranes were washed with PBST and then developed with an enhanced chemiluminescence kit (Thermo Scientific, USA). Images were captured by Tanon developer(Tanon Finedo X6, China). Image J(version 1.53t, USA) was used to analyze the protein bands.

## Quantitative real-time PCR

Total RNA was extracted using TRIzol reagent (Invitrogen, USA) following the protocol supplied by the manufacturer. One microgram of total RNA was reverse-transcribed using a Reverse Transcription System (Bio-Rad, USA), then the cDNA was stored at -20˚C Amplification was conducted using StepOnePlusTM real-time PCR system (ABI, USA). Reverse transcription was performed in a final volume of 20 μL containing 3 μL cDNA, 0.4 μL forward and reverse primer (10uM), 10 μL 2×SYBR Green qPCR Master Mix, and 6.2 μL RNase free water. The temperature cycling conditions were 40 cycles, including denaturation at 95˚C for 30 s, primer annealing for 30s at 60˚C, and primer extension at 95˚C for 15s. The mRNA of GAPDH served as an internal reference gene. The relative expression of genes was analyzed using the $2^{-\Delta\Delta Ct}$ method. The sequences of the following primers used for Slc18a2, Cox7a2l, Slc6a3, Psmd4, Nfe2l2, Psmd9, Mfn2, HO-1, GCLM, NQO1, GCLM, GAPDH qPCR are shown in S1 and S2 Tables.

## RNA-seq

TRIzol reagent from Service Biotech was used for SN total RNA extraction. The RNA samples were subjected to quantification and integrity analysis using a volume of 1 μl per sample on

the Agilent 2100 Bioanalyzer. The production of RNA samples involved the utilization of 3 μg of total RNA per sample. Following this, the RNA substance was subjected to 50-base pair single-end sequencing with a BGISEQ-500 sequencer. A minimum of 20 million clean sequencing reads were obtained from each sample. The differential expression analysis of two groups was conducted using the DESeq2 R package (version 1.42.0), with two biological replicates per condition.DESeq2 offers statistical algorithms for identifying differential expression in digital gene expression data, utilizing a model that follows the negative binomial distribution. The Benjamini and Hochberg technique was employed to control the false discovery rate and make adjustments for the obtained P values. Genes that were identified as differentially expressed by DESeq and had an adjusted P value less than 0.05 were labeled as such. Differentially expressed genes (DEGs) were operationally defined as genes exhibiting a fold change ratio (FDR) below 0.01 and a log2 fold change exceeding 1 (indicating upregulation) or falling below -1 (indicating downregulation).PMID: 25516281. The study employed the Gene Ontology (GO) and pathway annotation, along with enrichment analysis, utilizing the Gene Ontology Database (http://www.geneontology.org) and the KEGG pathway database (http://www.genomejp/kegg). The Metascape website was used to analyze the differences in gene expression between the MPTP vs CON, MPTP_BFT-vs-MPTP, and MPTP_MCC950-vs-MPTP comparison groups. The identification of upstream transcription factors that play a central role was achieved using a comparative analysis utilizing the TRRUST transcription factor database (https://metascape.org/gp/index.html).

## Determination of SOD, GSH, and MDA

Midbrain substantia nigra tissue was thawed at room temperature and blotted on filter paper. About 200 mg of brain tissue was taken and washed with pre-cooled saline. A homogenizer was filled with 9 times the mass of pre-cooled saline, which was converted to 100 g/l of brain homogenate. The homogenate was centrifuged at 3500 r/min for 15 min at 4˚C. An appropriate amount of supernatant was taken for tissue protein quantification, and the contents of SOD, GSH, and MDA were determined according to the specifications and usage of SOD, GSH and MDA kits (Abbkine, USA).

## High performance liquid chromatography (HPLC)

High-performance liquid chromatography (HPLC) was utilized to identify neurotransmitters, such as DA and its metabolites, in brain tissue. The experimental procedure entailed the measurement and sonication of the striatum tissue using ice-cold 0.01 mM perchloric acid solution containing 0.01% EDTA. This was subsequently followed by centrifugation and filtration. A mobile phase was employed for separation at a flow rate of 1.2 mL/min. The mobile phase consisted of 85 mM citric acid, 100 mM anhydrous sodium acetate, 0.2 mM disodium ethylenediaminetetraacetic acid (EDTA-2Na), and 15% (v/v) methanol (pH 3.68). The utilization of standard curves or the internal standard approach facilitated the determination of DA and its metabolite concentrations in ng/mg tissue weight by the examination of the chromatograms.

## Statistical analysis

The data is presented in the form of the means±SEM, utilizing a minimum of three biologically independent studies. A minimum of three biologically separate tests had comparable outcomes, from which representative morphological photos were captured. Statistical significance was assessed through the utilization of an unpaired two-tailed Student t-test, either one-way or two-way ANOVA. Subsequently, a multiple comparison test, either Tukey's or Bonferroni's, was conducted to compare the various treatment groups. The data were subjected to statistical

analysis using the designated procedures outlined in the figure captions, utilizing the SPSS program (version 22.0). Significance was deemed acceptable at p-values of 0.05, 0.01, or 0.001.

## Results

### Effect of BFT on body weight in MPTP-induced PD mice

All mice's body weights were measured concurrently during the medication delivery process. The mice in the model group saw a loss in body weight starting with the injection of MPTP on day 1, which was lower than that of the control group. Their body weight progressively returned to normal levels following the cessation of MPTP injections, as seen in Fig 1A. The mice in the BFT intervention group exhibited varying increases in body weight compared to the model group, but the changes were not statistically significant. After the trial, the body weights of mice in both the model group and the BFT intervention group had reverted to their usual levels.

### Effect of BFT on the behavioral properties of mice with MPTP-induced PD

**Pole test.** Fig 1B shows that the T-Turn time of the MPTP group rose by 27.1% compared to the Control group, whereas the T-Turn time of the MPTP+BFT group reduced by 18.3% compared to the MPTP group. The T-Total time of the MPTP+BFT group dropped by 18.3% to $8.19 \pm 0.57$ seconds (P value). An 18.3% increase in T-Total time was observed in the MPTP group compared to the Control group, with a statistically significant P value of 0.0069 ($< 0.01$). Fig 1C shows that the T-Total time of the MPTP group increased by 27.1% ($P < 0.0001$) compared to the Control group, and the T-Total time of the MPTP+BFT group increased by 27.1% ($P < 0.0001$) compared to the MPTP group. The total duration was reduced by 14.0% in the MPTP+BFT group ($6.88 \pm 0.51$ s) compared to the MPTP group ($8.06 \pm 0.43$ s), with a P value of 0.0209 ($P < 0.05$).

**Hang test.** Fig 1D displays that the average hanging time of PD model mice in the MPTP group was ($41.1 \pm 1.53$) s, significantly lower than the Control group ($50.1 \pm 1.53$) s, with a p-value of 0.0003 ($P < 0.001$). The difference was statistically significant. In contrast, the hanging time of mice in the MPTP+BFT intervention group ($47.6 \pm 1.54$) s increased compared to the MPTP group ($41.1 \pm 1.53$) s, with a p-value of 0.0109 ($P < 0.05$), also statistically significant. In the MPTP group, the time increased from ($41.1 \pm 1.53$) s to 1.54 s, with a statistically significant p-value of 0.0109 ($P < 0.05$).

**Step experiment.** Fig 1E illustrates that the Control group mice left consistent and straight trails, while the MPTP-induced PD mice left trails of varying lengths and directions. The trails of the MPTP+BFT group were similar to those of the Control group. Fig 1F displays a comparison of step length results among different groups. The Control group had an average step length of ($6.56 \pm 0.16$) cm, which was significantly longer than the average step length of PD model mice induced by the MPTP group ($4.55 \pm 0.25$) cm, with a P value of less than 0.0001. Additionally, the average step length of mice in the MPTP+BFT intervention group ($5.79 \pm 0.21$) cm was greater than that of the MPTP group ($4.55 \pm 0.25$) cm. In the MPTP group, the measurement was larger than 4.55 cm with a margin of error of $\pm 0.25$ cm. The difference was statistically significant with a P value of 0.0016 ($P < 0.01$).

**Open field test.** Fig 1G displays that the overall crawling distance in the MPTP-induced PD model group fell by 35.8% compared to the Control group, with statistical significance ($P < 0.0001$). Additionally, compared to the MPTP group, the MPTP + BFT group showed. . . In the intervention group, the total distance crawled rose by 23.59% to $2196.31 \pm 86.24$ cm, with a significant p-value of 0.0014, $P < 0.01$. In Fig 1H, the MPTP group had 39.11% fewer line traversals ($164.87 \pm 9.70$) compared to the Control group ($270.87 \pm 8.13$), $P < 0.0001$. The

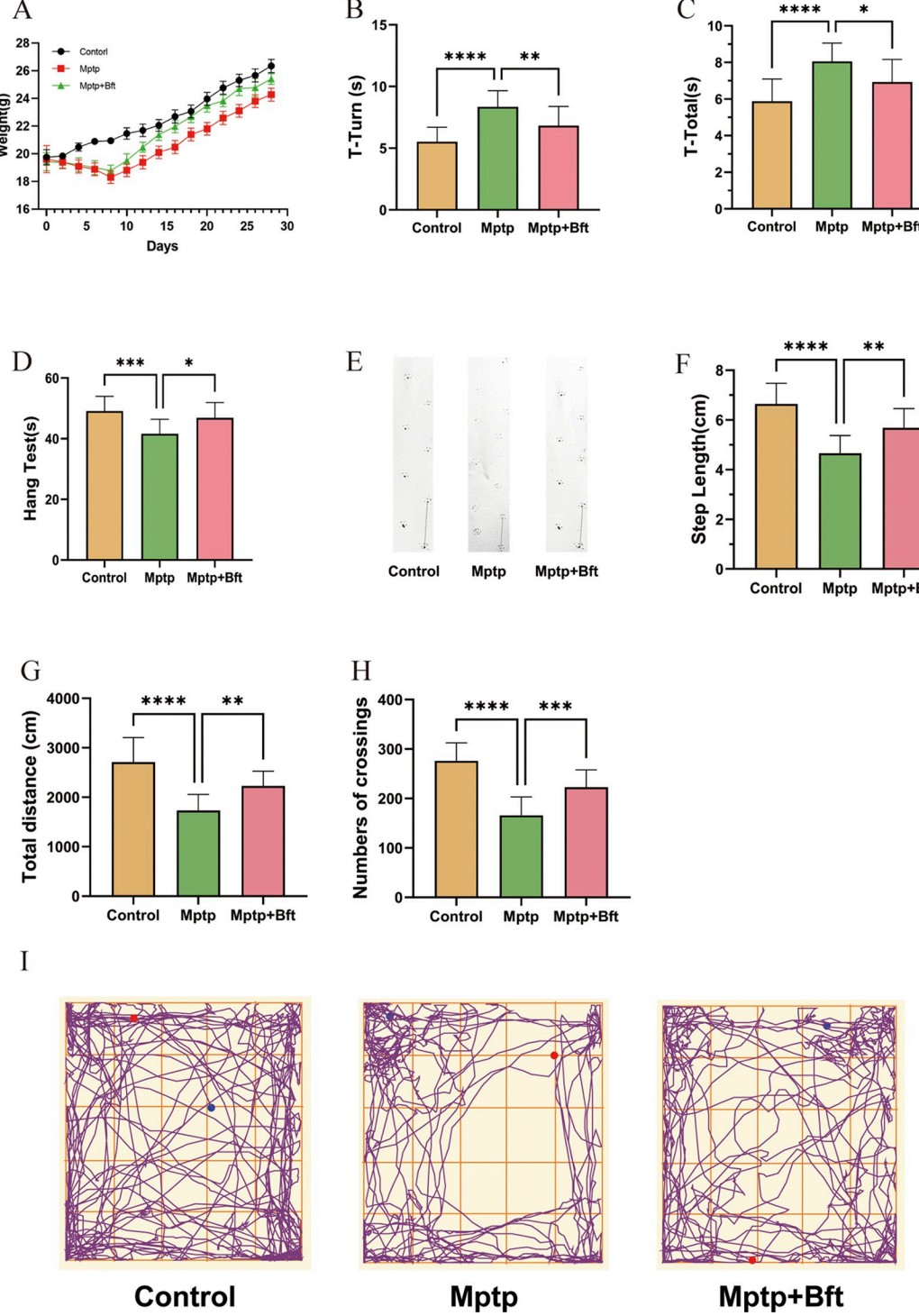

**Fig 1. Effect of BFT on the behavioral properties of mice with MPTP-induced PD.** (**A**) Body weight changes in three groups of mice over a period of 1 months (n = 15 animals for each group). (**B, C**) Pole test plots of three groups of mice. The duration from when the mouse began moving to when its head completed the turn downward (T-Turn) and the overall time from when the mouse started standing on the ball to when it reached the bottom of the pole (T-Total) was noted. (**D**) The Hang test plot of three groups of mice. (**E,F**) Plots of the double hind footprints of three groups of mice, and the length of the right rear footprint line is the STEP LENGTH. (**G,H**) VisuTrack Animal Behavior Analysis Software was utilized for 5 minutes to track the mice's total crawling distance and line crossings. (**I**) The mouse crawling trajectory diagram. (Behavioral tests, Data are expressed as mean ± SEM, n = 15, *P<0.05, **P<0.01, ***P<0.001, ****P<0.0001).

MPTP+BFT group had a 27.39% increase in line traversals (227.06 ± 10.30) compared to the MPTP group (164.87 ± 9.70), P < 0.001. Fig 1I demonstrates that the Control and MPTP+BFT groups followed comparable paths, however, the MPTP group had a much shorter trajectory length and passed the center of the field less frequently.

## BFT treatment attenuates dopaminergic neurotoxicity in MPTP-induced PD model mice

**Midbrain substantia nigra and striatum TH immunofluorescence staining result.** Fig 2A illustrates that the nigrostriatal DA neurons in the Control group appeared healthy with well-organized axons and dendrites forming a distinct meshwork. In contrast, the nigrostriatal DA neurons in the MPTP group were sparse, and disorganized, and exhibited blurred cytosolic contours, varied morphologies, wrinkled cytosol, pale cytoplasm, and significant loss of axons and dendrites. The MPTP+BFT group showed a considerable increase in the number of nigrostriatal DA neurons, which were densely dispersed, cleanly aligned, and had a fuller morphology. The neuronal cells had a distinct cytoplasm outline that was typically spherical or oval. Fig 2C displayed alterations in the quantity of TH+ cells in the substantia nigra of mice. The number of TH+ cells in the substantia nigra of the MPTP group was significantly lower compared to the Control group (22.83±12.45 cells in the MPTP group vs. 39.17±11.88 cells in the Control group; P < 0.05). Conversely, the number of TH+ cells in the MPTP+BFT group was significantly higher than in the MPTP group, and the number of TH+ cells in the BFT group was also significantly increased. The number of TH+ cells considerably increased with a P value of < 0.05 (22.83±12.45 in the MPTP group vs. 37.50±7.85 in the MPTP + BFT group).

Fig 2B & 2D illustrates that the density of TH-positive fibers was significantly reduced in the MPTP group compared to the Control group (0.47 ± 0.23 in the MPTP group vs. 0.98 ± 0.32 in the Control group), due to MPTP-induced dopaminergic neuronal toxicity. Conversely, the density of TH-positive fibers was significantly higher in the MPTP+BFT group compared to the MPTP group (0.47±0.23 vs. 0.85±0.15 in the MPTP group vs. 0.85±0.15 in the MPTP+BFT group). The data indicate that BFT intervention effectively restores the density of TH-positive fibers in the striatum.

**Nissl staining of the midbrain substantia nigra.** The structure of nigrostriatal neurons in the Control group mice, as depicted in Fig 2F, displayed clear characteristics. The cell bodies appeared round or oval, with distinct nucleoli and nuclear membranes. The nucleoli were centrally located, and the cytoplasm contained blue microsomes in a uniform color shade. The MPTP-induced PD model mice group exhibited structural abnormalities such as disappearing nucleoli, fuzzy nuclear membrane edges, uneven staining, reduced cytosol size, widened cytosol gap, deeply stained cytoplasm, and a significant decrease in the number of Nissl bodies and nerve cells. In the MPTP+BFT intervention group, the nucleus accumbens and nuclear membrane of the nigrostriatal neurons showed increased visibility compared to the MPTP group. The cytosol seemed somewhat enlarged, with the cytoplasm displaying light blue microsomes and a rather uniform staining pattern. Protecting BFT may enhance the quantity of Nissl-positive neurons. Fig 2E shows a significant decrease in the number of neurons in the substantia nigra of the midbrain of mice in the MPTP group (0.53±0.06) compared to the Control group, P < 0.0001. The number of neurons in the substantia nigra of the midbrain of mice in the MPTP+BFT intervention group (0.84±0.02) significantly increased compared to the MPTP group, P < 0.0001.

**BFT restores brain nigral and striatal TH expression in MPTP-induced PD models.** Fig 2G–2I displays the results indicating a significant decrease in TH expression in the midbrain substantia nigra and striatum of the MPTP group compared to the Control group

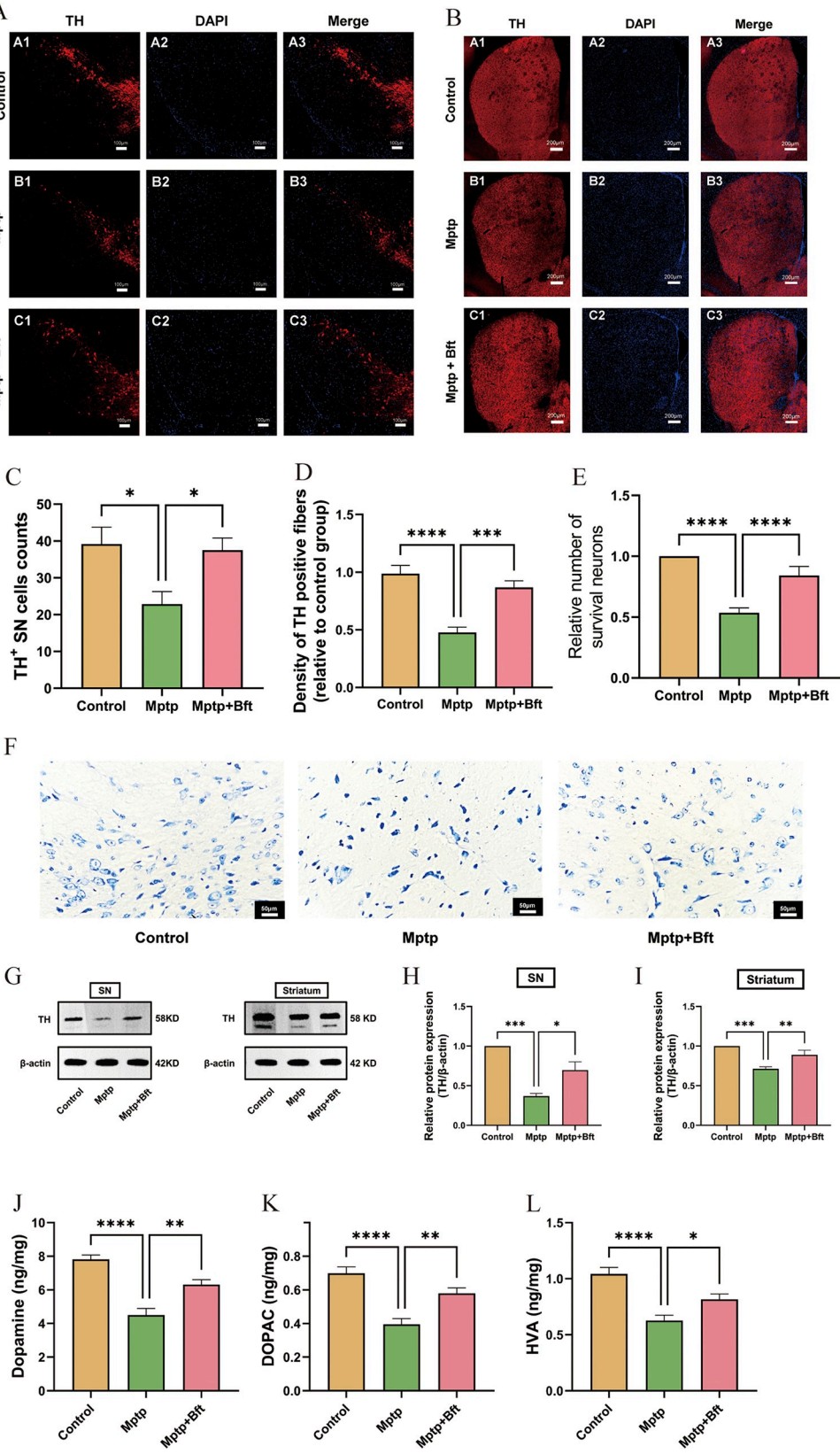

**Fig 2. BFT treatment attenuates dopaminergic neurotoxicity in MPTP-induced PD model mice and partially restores DA and metabolite levels in the striatum of PD mice.** (**A**) Results of immunofluorescence staining of nigral TH. (**B**) Results of immunofluorescence staining of striatal TH. (**C**) Results of cell counting of nigral TH⁺. (**D**) Statistical results of striatal TH⁺ fiber density. (**E**) Results of statistical analysis of the number of neurons in the midbrain substantia nigra assessed by Nissl staining. (**F**) Results of Nissl staining of the mouse midbrain substantia nigra. (**G**) WB protein bands for TH in the midbrain substantia nigra and striatum. (**H,I**) Mean gray scale analysis for quantification of TH and β-actin proteins. (**J,K&L**) Changes in the DA and intermediate metabolite:DOPAC&HVA levels in the striatum of MPTP-induced PD mouse models with BFT intervention. (Data are expressed as mean ± SEM, n = 6, *P<0.05, **P<0.01, ***P<0.001, ****P<0.0001).

(P < 0.01). Conversely, TH protein expression was notably enhanced in the BFT+MPTP intervention group compared to the MPTP group (P < 0.05).

**BFT partially restores DA and metabolite levels in the striatum of PD mice.** The results are shown in Fig 2J–2L. The DA, HVA, and DOPAC contents in the model group were significantly lower than those in the blank group (P < 0.05), with a decrease of 42.3%, 39.9%, and 43.4%, respectively; and the contents of DA, HVA, and DOPAC in the BFT intervention group were significantly higher than those in the model group (P < 0.05), with an increase of 40.3%, 30.0%, and 46.8%, respectively.

## RNA-seq analysis reveals gene expression patterns regulated by BFT and MCC950

**Differentially expressed genes and pathway analysis.** Fig 3A shows that there were 171 DEGs in MPTP versus CON, 164 DEGs in MPTP_BFT vs MPTP, and 177 DEGs in MPTP_MCC950 vs MPTP comparison groups. KEGG Pathway enrichment analysis showed that differentially expressed genes (DEGs) in the MPTP versus CON comparison were highly enriched in pathways such as TNF signaling, cytokine-cytokine receptor interaction, and parkinson's disease pathway. Differentially expressed genes in the comparison between MPTP_BFT and MPTP were shown to be enriched in parkinson's disease, dopamine synapse, TNF signaling, and other pathways. The NOD-like receptor signaling pathway was the most enriched route in the comparison between MPTP_MCC950 and MPTP, as seen in Fig 3B–3D. There were 76 differentially expressed genes (DEGs) shared between MPTP vs CON and MPTP_BFT vs MPTP, and 39 DEGs shared between MPTP vs CON and MPTP_MCC950 vs MPTP. The PD pathway showed the highest level of enrichment among the 76 shared DEGs. The primary enriched route for the 39 shared Differentially Expressed Genes (DEGs) was the NOD-like receptor signaling pathway, seen in S2 Fig.

**Transcription factor analysis and qPCR validation.** We conducted a transcription factor analysis on the Differentially Expressed Genes (DEGs) from comparisons between MPTP versus CON, MPTP_BFT vs MPTP, and MPTP_MCC950 vs MPTP to investigate the main upstream transcription factors associated with the mechanisms of BFT and MCC950. This study was carried out using the web program Metascape. The findings showed that Nfkb1 had a central regulatory function in the Differentially Expressed Genes (DEGs) across all three comparison groups. Nfkb1 had a greater regulatory impact on the differentially expressed genes (DEGs) in the MPTP vs CON and MPTP_MCC950 vs MPTP groups. Meanwhile, Nfe2l2 had a more noticeable regulatory influence on the DEGs in the MPTP_BFT vs MPTP group (Fig 3E).

Seven major Differentially Expressed Genes (DEGs)—Slc18a2, Cox7a2l, Slc6a3, Psmd4, Nfe2l2, Psmd9, and Mfn2—associated with the neuroprotective properties of BFT were chosen for validation by qPCR analysis to confirm the RNA-seq findings. The mRNA expression of these genes in the MPTP group was considerably lower compared to the control group

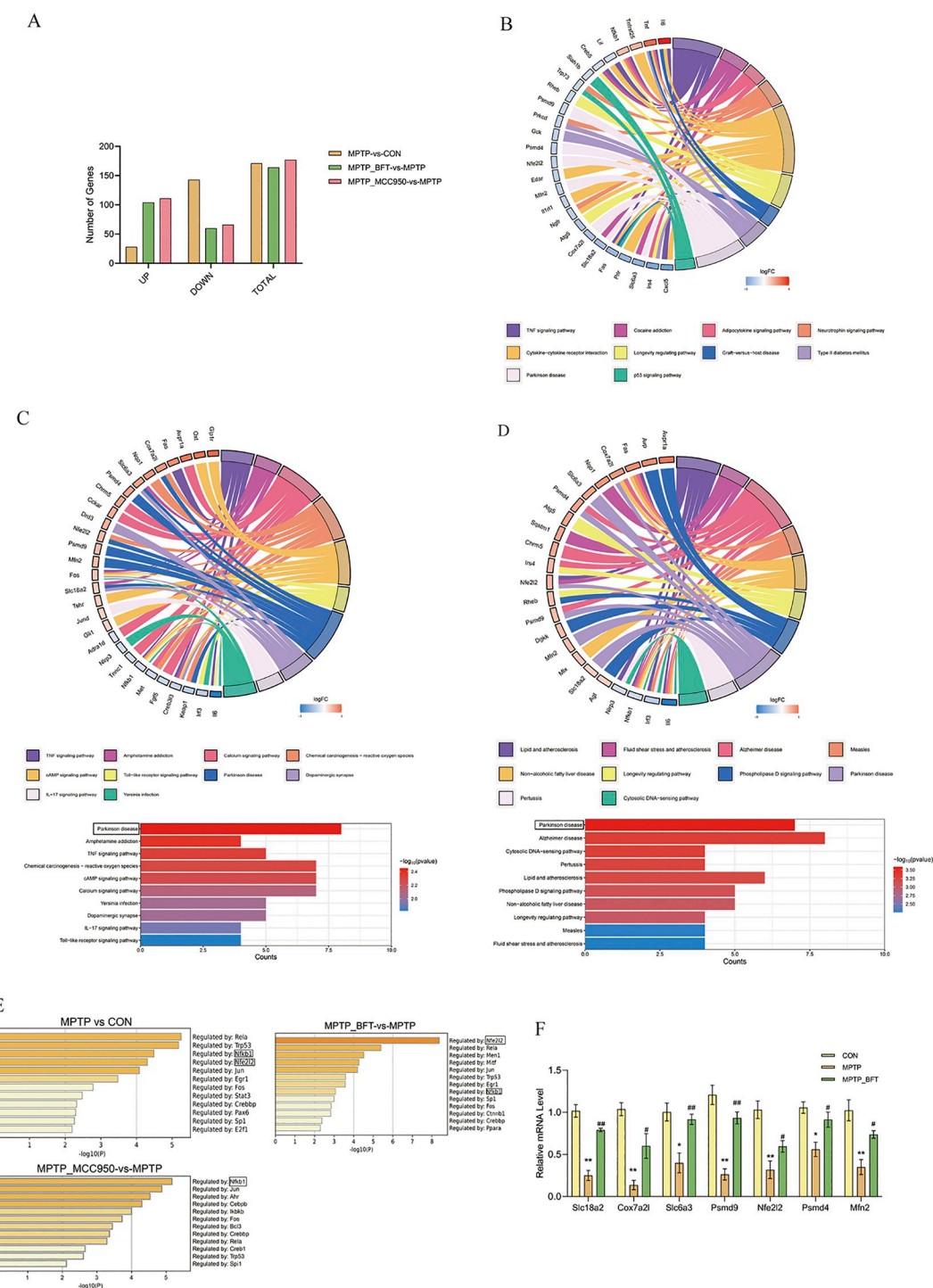

**Fig 3. RNA-seq analysis reveals gene expression patterns regulated by BFT and MCC950.** (**A**) Bar chart of differentially expressed genes (DEGs). The X-axis represents the categories of upregulated, downregulated, and total genes, while the Y-axis represents the corresponding number of DEGs. (**B**) KEGG pathway enrichment chord plot (MPTP-vs-CON). (**C**) KEGG enrichment analysis plot (MPTP_BFT vs MPTP). (**D**) Enrichment analysis of differentially expressed genes in the intersecting region of MPTP vs CON and MPTP_BFT vs MPTP comparison groups. String plots showing the pathways enriched for differential genes and the relative expression abundance of the differential genes associated with each pathway. Red: up-regulated genes, blue: down-regulated genes. Differential gene enrichment of KEGG signaling pathways, with -logl0 (P value) defining the top 10 significantly enriched KEGG pathways. (**E**) Summary of transcription factor enrichment analysis in Metascape (TRRUST). Upstream regulatory transcription factors of differentially expressed genes in each

comparison group were analyzed, defining the enrichment level as -log10(P value). A higher value indicates a more critical regulatory role, the greater the value, the more crucial the regulation. (**F**) qPCR (quantitative real-time polymerase chain reaction) validation of key genes. After BFT intervention, there were relative mRNA level changes in key signaling pathways related to differential genes. n = 3, MPTP vs CON, **P$<$0.01 and *P$<$0.05, MPTP_BFT vs MPTP, ## P$<$0.01 and # P$<$0.05.

(P $<$ 0.05). Following the BFT intervention, their expression significantly increased (P $<$ 0.05), almost reaching normal levels (Fig 3F).

## BFT activates Nrf2 pathway and improves antioxidant status in substantia nigra and striatum

**BFT activates the mRNA expression of downstream genes of Nrf2 in the substantia nigra.** The mRNA levels of downstream pathways associated with Nrf2 were assessed. According to the qPCR results presented in Fig 4A–4C, the mRNA expression of HO-1, GCLM, and NQO1 in the substantia nigra of MPTP-induced PD model mice decreased compared to the control group (P $<$ 0.05). In contrast, the mRNA expression of HO-1, GCLM, and NQO1 in the MPTP+BFT (200mg/kg) and MPTP+BFT (250mg/kg) intervention groups were upregulated compared to the model group, and these differences were statistically significant (P $<$ 0.05).

**BFT activates nuclear translocation of Nrf2 in the substantia nigra and striatum, as well as the expression of downstream proteins.** The western blotting results showed that MPTP caused a decrease in nuclear Nrf2 and an increase in cytoplasmic Nrf2 in both the substantia nigra and striatum, suggesting a disruption in Nrf2 nuclear translocation (Fig 4D). Nuclear Nrf2 levels rose but cytoplasmic Nrf2 levels declined in both MPTP+BFT groups relative to the MPTP group, indicating that BFT facilitated Nrf2 nuclear translocation. Furthermore, the protein levels of HO-1, GCLM, and NQO1 were increased to varying degrees in the MPTP+BFT groups compared to the MPTP group (Fig 4E). The data indicate that BFT may trigger Nrf2 signaling and subsequent antioxidant protein production in the mouse midbrain, perhaps leading to its neuroprotective benefits in the MPTP-induced PD model.

**BFT improves antioxidant enzyme activities and reduces oxidative damage in substantia nigra and striatum.** We assessed the antioxidant levels in the substantia nigra and striatum by evaluating SOD activity, MDA levels, and GSH activity. Fig 5 demonstrates that MPTP led to reduced SOD and GSH activity and higher MDA levels in both brain areas compared to the control group (P $<$ 0.05), suggesting oxidative stress. BFT therapy significantly elevated SOD and GSH activities while decreasing MDA levels compared to the MPTP group (P $<$ 0.05). The results show that BFT has the potential to enhance the antioxidant abilities of the substantia nigra and striatum in a mouse model of PD caused by MPTP.

## Discussion

PD is the second most common neurodegenerative disorder globally. According to data from 2016, the global prevalence of PD increased by 21.7% compared to 1990 [25, 26]. In 1817, Parkinson first published a monograph on "shaking palsy (an early term for Parkinson's Disease)" providing the initial description of this neurological disorder [27]. Since then, an increasing number of researchers have engaged in clinical and pathological studies on PD, revealing that its clinical manifestations primarily involve motor and non-motor symptoms, with pathological changes predominantly affecting the substantia nigra and striatum, including the formation of Lewy bodies (LBDs) [28, 29]. Based on these findings, dopamine replacement therapy and Deep Brain Stimulation (DBS) have been applied, offering partial improvement in PD symptoms. DBS is a surgical treatment that involves implanting electrodes into specific areas of the

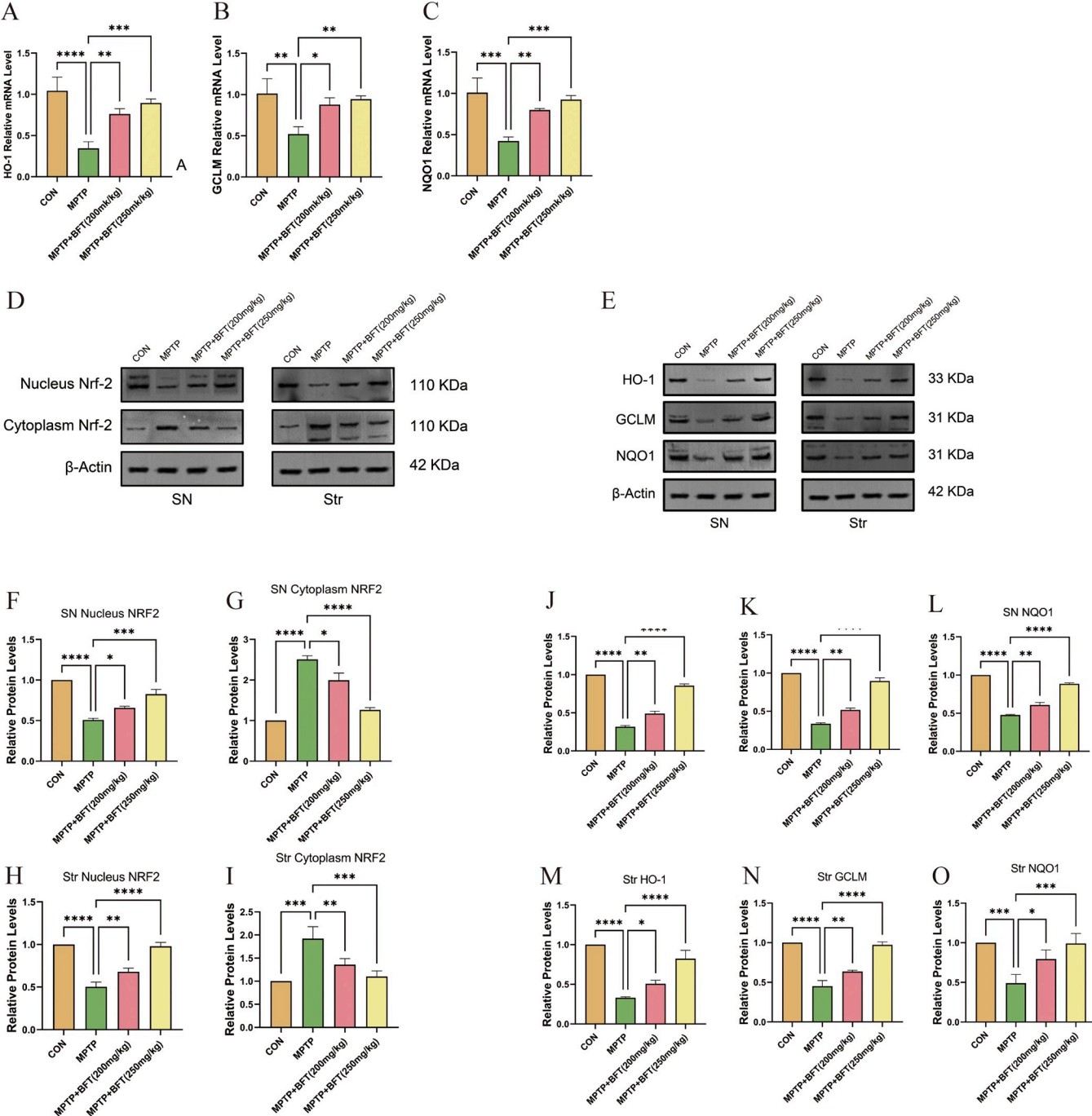

**Fig 4. BFT activates Nrf2 pathway in substantia nigra and striatum.** (**A,B&C**) mRNA expression of brain substantia nigra HO-1, GCLM, and NQO1 in various subgroups. (**D&F-I**) Nuclear and cytoplasmic protein expression of Nrf2 in substantia nigra and striatum of each group. D: Protein expression profiles of nuclear and cytoplasmic Nrf2, F-I: Relative expression levels of nuclear and cytoplasmic Nrf2 proteins, SN: Substantia Nigra, Str: Striatum. (**E&J-O**) Expression of Nrf2 downstream proteins in substantia nigra and striatum of each group. E: Protein expression profiles of downstream proteins of Nrf2 in the substantia nigra and striatum, J-O: Relative expression levels of downstream proteins of Nrf2 in the substantia nigra and striatum, SN: Substantia Nigra, Str: Striatum, *P<0.05, **P<0.01, ***P<0.001, ****P<0.0001.

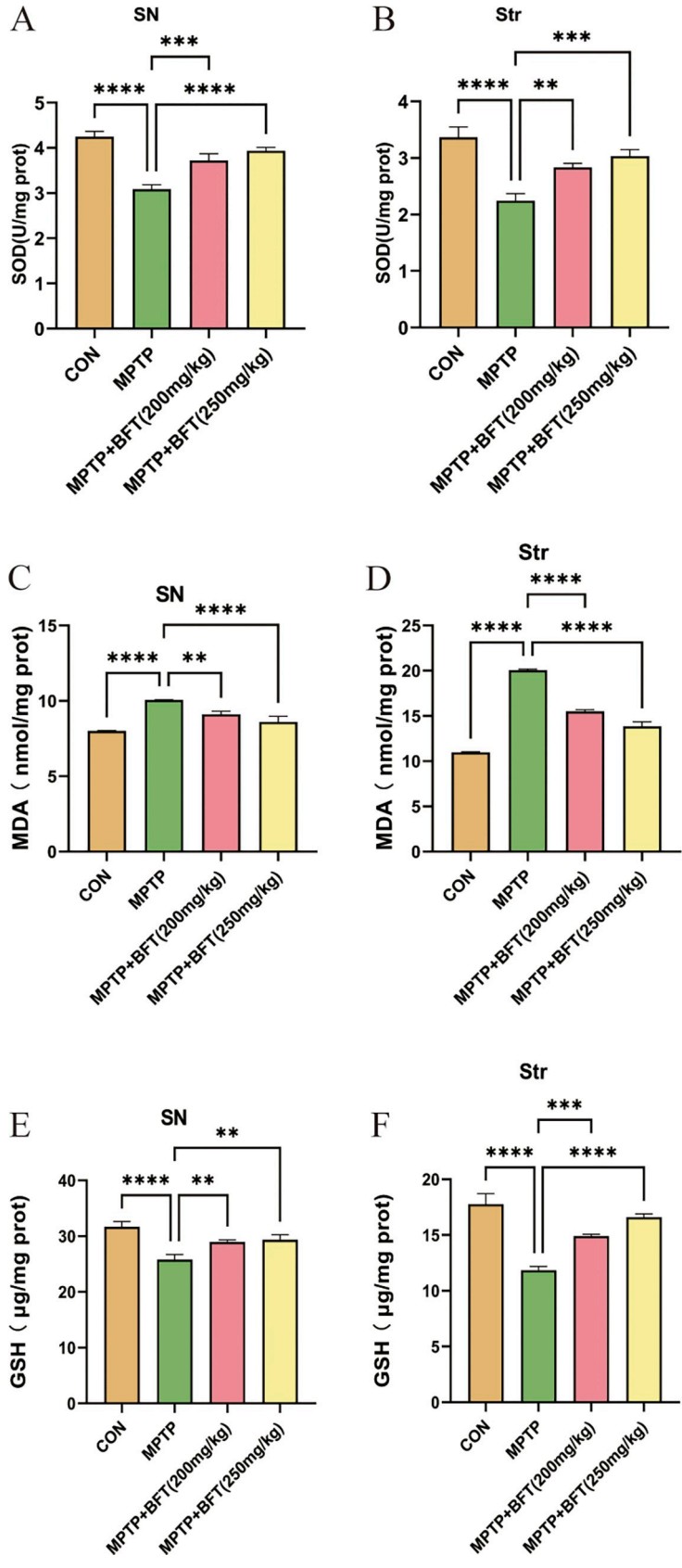

**Fig 5. BFT improves antioxidant enzyme activities and reduces oxidative damage in substantia nigra and striatum.** (**A,B**) Effect of BFT treatment on SOD activity in the substantia nigra of the midbrain and striatum of MPTP model group mice. (**C,D**) Effect of BFT treatment on MDA content in the substantia nigra of the midbrain and striatum of MPTP model group mice. (**E,F**) Effect of BFT treatment on GSH activity in the substantia nigra of the midbrain and striatum of MPTP model group mice. SN: Substantia Nigra, Str: Striatum, n = 3, *P<0.05, **P<0.01, ***P<0.001, ****P<0.0001.

brain to help manage symptoms of neurological disorders. However, these treatments do not halt the progression of the disease, and the use of medications often leads to various side effects and complications such as motor fluctuations and dyskinesias, causing significant distress to patients [30]. Clinical application of drugs like levodopa can moderately alleviate symptoms, but long-term use tends to result in diminishing efficacy and numerous side effects [4, 31].

Recent studies have underscored the therapeutic potential of natural compounds in PD. Notably, Ursolic acid, Withania somnifera, and Mucuna pruriens have demonstrated neuro-protection in PD models, mitigating oxidative stress, inflammation, and cell death pathways, thus alleviating neurological impairments [32–34]. Thiamine is a natural compound, and BFT is an organic compound containing a thiol group and represents a derivative of thiamine [35]. In comparison to thiamine, BFT features an open thiazole ring, facilitating its passage through cell membranes with enhanced permeability. This property allows BFT to traverse the blood-brain barrier, resulting in significantly higher bioavailability compared to thiamine [10–14]. Developed in Japan in the late 1950s, BFT initially found application in the treatment of dia-betic polyneuropathy [36, 37]. In recent years, BFT's role in the nervous system has gained attention, with studies indicating its potential to protect neurons from damage caused by ischemia, trauma, toxins, and other factors [38, 39]. Clinical research has demonstrated the safety and efficacy of BFT in improving cognitive function in mild Alzheimer's disease [19, 40]. Alzheimer's disease and PD are both neurodegenerative disorders characterized by similar neuronal damage and pathological changes [41]. Recent studies have discovered that BFT exhibits neuroprotective effects in a rat model of PD, although the underlying mechanisms remain unreported [15].

MPTP (1-methyl-4-phenyl-1,2,3,6-tetrahydropyridine) is a well-known toxin capable of selectively damaging dopaminergic neurons in the substantia nigra [16]. The MPTP-induced mouse model of PD is currently one of the most widely employed animal models for PD, pro-viding a robust simulation of the clinical and pathological features of the disease [42]. This study involved administering various dosages of BFT as an intervention therapy in the MPTP mouse model of PD to assess its protective effects on motor function and dopaminergic neu-rons in the animals. Pole test and Open Field Test is a useful method for evaluating the mouse movement disorder caused by striatal dopamine depletion [43, 44]. The study results showed that BFT considerably enhanced the performance of MPTP mice in several motor function tests, suggesting that BFT can effectively treat MPTP-induced motor impairments, this is con-sistent with previous research findings [15]. The potential therapeutic effects of BFT on the MPTP-induced Parkinson's disease (PD) mouse model may encompass a multitude of biologi-cal processes, including direct protective actions on dopaminergic neurons, anti-inflammatory effects, and anti-apoptotic mechanisms. These processes synergistically contribute to the ame-lioration of motor dysfunction. IF and western blotting investigations showed that BFT treat-ment led to an increase in the number of TH-positive dopaminergic neurons and TH protein expression levels in the substantia nigra and striatum of MPTP animals. Furthermore, BFT was able to partially replenish dopamine and its metabolite levels in the striatum of MPTP ani-mals. Collectively, our results show that BFT can safeguard dopaminergic neurons and prevent MPTP-induced neurotoxicity.

MCC950 is a specific inhibitor of the NOD-like receptor pyrin domain-containing protein 3 (NLRP3) [45]. Studies have demonstrated that MCC950 protects against nigral damage generated by MPTP in a PD model [46]. To further investigate the mechanisms by which BFT exerts neuroprotection on dopaminergic neurons, this study established an MCC950+MPTP control group. The impact of BFT and MCC950 on the transcriptome expression was analyzed using RNA sequencing technology, providing in-depth insights into their potential mechanisms in protecting dopaminergic neurons. The enrichment analysis of differentially expressed genes revealed that BFT predominantly regulates signal pathway: map05012 Parkinson disease, while MCC950 primarily targets the NOD-Like receptor pathway. This finding suggests that these two drugs may operate through distinct protective mechanisms, and their combined application holds the potential for synergistic effects. Additional investigation using the TRRUST database on Metascape indicated that Nfe2l2 plays a crucial role as a regulatory transcription factor in the MPTP_BFT versus MPTP comparison group, suggesting that activating Nfe2l2 might be a beneficial approach for PD treatment. The Nfe2l2 gene encodes the Nrf2 protein, which is crucial in regulating oxidative stress [47].

BFT enhances cellular antioxidant capacity by up-regulating Nfe2l2 expression, facilitating its release from Keap1, and promoting Nrf2 nuclear translocation, leading to the activation of antioxidant genes like HO-1, GCLM, and NQO1. Enhancing the expression of antioxidant proteins improves the body's capacity to eliminate oxygen radicals and repair oxidative damage [48–50]. During the oxidative stress process, by inducing HO-1 expression, it can exert potent antioxidant, anti-inflammatory, and anti-apoptotic effects, which are likely due to HO-1 degrading heme and generating a series of cytoprotective products (ferrous ions, biliverdin, and carbon monoxide) [51] BFT increased SOD and GSH activities and decreased MDA levels in the midbrain substantia nigra and striatum, indicating its role in causing oxidative stress in these regions, demonstrating a notable antioxidant impact.

Although the study suggests that BFT's protective effect is closely related to the activation of the Nrf2 pathway, its regulation of mitochondrial function, reduction of inflammatory response, and other protective mechanisms need to be further explored. Moreover, the small sample size means that the generalizability of the conclusions needs further verification. In addition, the efficacy of BFT in other PD models such as the commonly used 6-hydroxydopamine (6-OHDA) model needs to be further explored. Future studies could explore optimal combination therapy through joint intervention with other drugs, such as Levodopa and Benserazide Hydrochlo-ride, to investigate synergistic efficacious regimens. At the same time, the sample size should be expanded and clinical trials should be conducted to screen for populations sensitive to BFT treatment according to patients' gene expression profiles. To advance BFT towards clinical application, well-designed clinical trials are necessary. As with any potential therapeutic, assessing the safety and tolerability of BFT in clinical settings is crucial. Studies should monitor for any adverse effects and determine the optimal dosage for human use.

## Conclusions

The research demonstrated that BFT enhanced motor function and safeguarded dopaminergic neurons in a mouse model of PD by stimulating the Nrf2-ARE signaling pathway and boosting the production of antioxidant enzymes. BFT increased the expression of important genes associated with PD that had been decreased by MPTP. The ongoing study expands the range of pharmacological uses for BFT and provides empirical evidence of its potential as a treatment for PD. Moreover, a comparison with MCC950 reveals different target areas for the two medicines, suggesting potential synergistic effects when combined. Our research emphasizes the crucial regulatory role of the transcription factor Nfkb1 in the MPTP-induced PD mouse

model. This suggests that targeting its excessive activation might be a new treatment strategy. It is important to recognize that these results are derived from animal experiments.

## Supporting information

**S1 Fig. Experimental dosing time diagram.** After a one-week acclimatization period, groups of mice received intraperitoneal injections of MPTP at a dose of 30 mg/kg for five consecutive days to induce a Parkinson's disease model. The control group received the same amount of saline solution containing 0.9% NaCl. Mice were orally administered BFT at a dose of 200 mg/kg and 250 mg/kg, respectively, for 28 days, while the other group received intraperitoneal injections of MCC950 at a dose of 10 mg/kg for the same period of time.
(TIF)

**S2 Fig. Heat maps, volcano and venn of differentially expressed genes. (A)** Differential gene expression profiles between MPTP and CON comparison groups. (Left) Heatmap of relative expression of 171 differential genes; (Right) Differential gene volcano plot. **(B)** Differential gene expression profiles between MPTP_BFT and MPTP comparison groups. (Left) Heatmap of relative expression of 164 differential genes; (Right) Differential gene volcano plot. **(C)** Differential gene expression profiles in the intersecting region of MPTP vs CON and MPTP_BFT vs MPTP comparison groups. (Left) Heatmap of relative expression of 76 differential genes; (Right) Differential gene volcano plot. (**D**) MPTP_MCC950-vs-MPTP Differential Gene Expression Profile. (Left) Heat map of relative expression of 177 differential genes; (Right) Differential gene volcano map. (**E**) Differential gene expression profiles of MPTP vs CON and MPTP_MCC950-vs-MPTP shared changes. (Left) Heatmap of expression abundance of 39 co-altered differential genes; (Right) MPTP vs CON &MPTP_MCC950-vs-MPTP co-altered differential gene Venn diagram. Note: Differential gene criteria: p<0.05 and |Log2FC|≥1. red: up-regulated genes; blue: down-regulated genes.
(TIF)

**S1 Table. Primer information table.**
(XLSX)

**S2 Table. Primer information table.**
(XLSX)

**S1 Raw images.**
(PDF)

## Acknowledgments

The authors of this paper express their sincere gratitude to Ms. Chen Na for her valuable assistance and support during the experimental phase of this study, which ensured the smooth progress of the experiments.

## Author Contributions

**Conceptualization:** Kai Wang, Lei Wang, Yu Wang.

**Data curation:** Kai Wang, Jinwei Yang, Yu Wang.

**Funding acquisition:** Huaiyu Li, Yu Wang.

**Methodology:** Chao Han, Wenhao Xu.

**Project administration:** Huaiyu Li.

**Resources:** Chao Han, Lei Wang.

**Software:** Kai Wang, Jinwei Yang, Wenhao Xu.

**Writing – original draft:** Kai Wang.

**Writing – review & editing:** Yu Wang.

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
