## [Decision Letter · Decision Letter 0]

2 Apr 2024

PONE-D-24-09739Benfotiamine Protects MPTP-Induced Parkinson's Disease Mouse Model via Activating Nrf2 Signaling PathwayPLOS ONE

Dear Dr. Wang,

Thank you for submitting your manuscript to PLOS ONE. After careful consideration, we feel that it has merit but does not fully meet PLOS ONE’s publication criteria as it currently stands. Therefore, we invite you to submit a revised version of the manuscript that addresses the points raised during the review process.

We look forward to receiving your revised manuscript.

Kind regards,

Sachchida Nand Rai, Ph.D.

Academic Editor

PLOS ONE

Journal Requirements:

"This study was supported by the National Natural Science Foundation of China (82071460) and Science and Technology Major Projects in Anhui Province (202103a07020015)."

"Acknowledgments: This study was supported by the National Natural Science Foundation of China (82071460) and Science and Technology Major Projects in Anhui Province (202103a07020015)."

Please remove any funding-related text from the manuscript.

4. In the online submission form you indicate that your data is not available for proprietary reasons and have provided a contact point for accessing this data. Please note that your current contact point is a co-author on this manuscript. According to our Data Policy, the contact point must not be an author on the manuscript and must be an institutional contact, ideally not an individual. Please revise your data statement to a non-author institutional point of contact, such as a data access or ethics committee, and send this to us via return email. Please also include contact information for the third party organization, and please include the full citation of where the data can be found.

8. Please include a separate caption for each figure in your manuscript.

9. We notice that your supplementary tables S1 and S2 are uploaded with the file type 'Other'. Please amend the file type to 'Supporting Information'. Please ensure that each Supporting Information file has a legend listed in the manuscript after the references list.

**Additional Editor Comments:**

Dear Authors,

Kindly revise your manuscript as per the reviewer suggestions. In addition, highlights each and every changes in your manuscript.

Reviewers' comments:

Reviewer's Responses to Questions

**Comments to the Author**

1. Is the manuscript technically sound, and do the data support the conclusions?

Reviewer #1: Yes

Reviewer #2: Yes

2. Has the statistical analysis been performed appropriately and rigorously? 

Reviewer #1: Yes

Reviewer #2: Yes

3. Have the authors made all data underlying the findings in their manuscript fully available?

Reviewer #1: No

Reviewer #2: Yes

4. Is the manuscript presented in an intelligible fashion and written in standard English?

Reviewer #1: Yes

Reviewer #2: No

5. Review Comments to the Author

Reviewer #1: Authors Comment

Author had written a good article entitled as “Benfotiamine Protects MPTP-Induced Parkinson's Disease Mouse Model via Activating Nrf2 Signaling Pathway”. Author aimed to examined the potential protective effects of BFT against damage to dopamine neurons in a PD animal model, as well as the underlying mechanisms. It is nicely written however, I have certain concern which, I have listed below:

1.Please mention which dose (200 or 250 mg/kg) of Benfotiamin (BFT), author has used in the experiment.

2.Figure S1 only showed the BFT 200 mg/kg response on mice. This makes incomplete experimentation.

3.If possible, please elaborate and pictorially depict the 200 mg/kg and 250 mg/kg, and MCC950 intraperitoneal injections, dosing response on mice in each experimentation (either on behavioral test or on molecular experimentation).

4.Please, elaborate BFT dose used in the present manuscript and why author has chosen the dose.

5.The present manuscript showed that mice in the MPTP group exhibited a loss in body weight upon MPTP injection, which gradually returned to normal levels after the cessation of MPTP injections. The mice in the BFT intervention group showed some increases in body weight compared to the MPTP group, but these changes were not statistically significant.

6.In the present study, the duration of BFT intervention in the study might not be sufficient to fully assess its long-term effects on PD progression. Longer intervention periods would be necessary to evaluate the sustained benefits of BFT treatment and its potential to slow down disease progression.

7.The present study suggests that BFT exerts its neuroprotective effects through various mechanisms such as Nrf2 pathway activation and antioxidant enhancement, the exact mechanisms underlying its action are not fully elucidated. Further research is needed to comprehensively understand how BFT functions at the molecular level and its potential interactions with other biological pathways.

8.If possible can author show, the degeneration of dopaminergic neurons by MPTP dosing and dopaminergic neuron regeneration by BFT using tyrosine hydroxylase staining.

Reviewer #2: My comments on the manuscript entitled “Benfotiamine Protects MPTP-Induced Parkinson's Disease Mouse Model via Activating Nrf2 Signaling Pathway” are as follows.

How does BFT intervention impact motor deficits in the PD mouse model, as assessed by the pole test, hang test, gait analysis, and open field test? Are there specific improvements noted in certain types of motor functions over others?

Can you elaborate on the methods used to evaluate the extent of damage to dopaminergic neurons in the substantia nigra and striatum following MPTP-induced injury, particularly regarding the techniques of Immunohistofluorescence, Nissl staining, and Western blot analysis of Tyrosine Hydroxylase (TH)?

What specific changes in dopamine (DA) levels and its metabolites were observed following BFT intervention, as measured by High Performance Liquid Chromatography (HPLC)? Were these changes statistically significant, and how do they correlate with the observed neuroprotective effects?

Regarding the RNA-seq and bioinformatics analysis, what were the key pathways and genes affected by BFT intervention in the substantia nigra tissues? How do these findings compare with those from treatment with the NLRP3 inhibitor MCC950, and what implications do these differences have for Parkinson's disease research?

Could you provide more detail on the findings related to the activation of the Nrf2 pathway by BFT intervention? Specifically, how was the movement of Nrf2 to the nucleus assessed, and what were the observed changes in the expression of downstream antioxidant genes and enzymes in the substantia nigra and striatum?

In what ways did BFT treatment affect oxidative damage markers such as MDA content, GSH activity, and SOD activity in the substantia nigra and striatum? Were these changes consistent with the observed neuroprotective effects, and do they provide further insight into the mechanisms of BFT's action in PD?

Given the findings of this study, what are the potential implications for the development of BFT as a therapeutic agent for Parkinson's disease in humans? What additional preclinical or clinical studies are needed to further investigate its efficacy and safety profile in PD patients?

How does the global prevalence of Parkinson's Disease (PD) in 2016 compare to that of 1990, according to the provided data?

What are the primary clinical manifestations of PD, and which brain regions are predominantly affected by its pathological changes?

What are the limitations of current treatments for PD, such as dopamine replacement therapy and Deep Brain Stimulation?

How does thiamine (vitamin B1) potentially influence the onset and development of PD, based on the described research findings?

What are the advantages of using benfotiamine (BFT) over thiamine in terms of its bioavailability and ability to traverse the blood-brain barrier?

Similar to Benfotiamine, Mucuna pruriens, Ursolic acid and Chlorogenic acid also exhibited similar kind of therapeutic activity in MPTP intoxicated mouse model. Cite original article for the same and correlate with your own outcomes.

Can you summarize the findings regarding the neuroprotective effects of BFT in both Alzheimer's disease and PD, as described in the text?

How does 1-methyl-4-phenyl-1,2,3,6-tetrahydropyridine (MPTP) induce damage in dopaminergic neurons, and what model of PD does it represent?

What were the main results of the study investigating the protective effects of BFT on motor function and dopaminergic neurons in the MPTP mouse model of PD?

What is MCC950, and how does it relate to the NOD-like receptor pyrin domain-containing protein 3 (NLRP3)?

How did the study utilize RNA sequencing technology to investigate the protective mechanisms of BFT and MCC950 on dopaminergic neurons, and what were the key findings regarding their potential synergistic effects?

Provide exact p value for all your histograms.

Positive control is missing in your experiment.

Provide 4x, 10x and other higher magnification images for all your immunoflourescence.

Provide complete western blot, not the cut one.

Improve clarity and resolution of all the western blot in your manuscript.

6. PLOS authors have the option to publish the peer review history of their article (what does this mean?). If published, this will include your full peer review and any attached files.

Reviewer #1: **Yes: **Dr Vineeta Singh

Reviewer #2: **Yes: **Payal Singh

---

## [Author Response · Author response to Decision Letter 0]

20 May 2024

Reviewer #1: Authors Comment

Author had written a good article entitled as “Benfotiamine Protects MPTP-Induced Parkinson's Disease Mouse Model via Activating Nrf2 Signaling Pathway”. Author aimed to examined the potential protective effects of BFT against damage to dopamine neurons in a PD animal model, as well as the underlying mechanisms. It is nicely written however, I have certain concern which, I have listed below:

1.Please mention which dose (200 or 250 mg/kg) of Benfotiamin (BFT), author has used in the experiment.

2.Figure S1 only showed the BFT 200 mg/kg response on mice. This makes incomplete experimentation.

3.If possible, please elaborate and pictorially depict the 200 mg/kg and 250 mg/kg, and MCC950 intraperitoneal injections, dosing response on mice in each experimentation (either on behavioral test or on molecular experimentation).

4.Please, elaborate BFT dose used in the present manuscript and why author has chosen the dose.

5.The present manuscript showed that mice in the MPTP group exhibited a loss in body weight upon MPTP injection, which gradually returned to normal levels after the cessation of MPTP injections. The mice in the BFT intervention group showed some increases in body weight compared to the MPTP group, but these changes were not statistically significant.

6.In the present study, the duration of BFT intervention in the study might not be sufficient to fully assess its long-term effects on PD progression. Longer intervention periods would be necessary to evaluate the sustained benefits of BFT treatment and its potential to slow down disease progression.

7.The present study suggests that BFT exerts its neuroprotective effects through various mechanisms such as Nrf2 pathway activation and antioxidant enhancement, the exact mechanisms underlying its action are not fully elucidated. Further research is needed to comprehensively understand how BFT functions at the molecular level and its potential interactions with other biological pathways.

8.If possible can author show, the degeneration of dopaminergic neurons by MPTP dosing and dopaminergic neuron regeneration by BFT using tyrosine hydroxylase staining.

 

Reviewer #2: My comments on the manuscript entitled “Benfotiamine Protects MPTP-Induced Parkinson's Disease Mouse Model via Activating Nrf2 Signaling Pathway” are as follows.

How does BFT intervention impact motor deficits in the PD mouse model, as assessed by the pole test, hang test, gait analysis, and open field test? Are there specific improvements noted in certain types of motor functions over others?

Can you elaborate on the methods used to evaluate the extent of damage to dopaminergic neurons in the substantia nigra and striatum following MPTP-induced injury, particularly regarding the techniques of Immunohistofluorescence, Nissl staining, and Western blot analysis of Tyrosine Hydroxylase (TH)?

What specific changes in dopamine (DA) levels and its metabolites were observed following BFT intervention, as measured by High Performance Liquid Chromatography (HPLC)? Were these changes statistically significant, and how do they correlate with the observed neuroprotective effects?

Regarding the RNA-seq and bioinformatics analysis, what were the key pathways and genes affected by BFT intervention in the substantia nigra tissues? How do these findings compare with those from treatment with the NLRP3 inhibitor MCC950, and what implications do these differences have for Parkinson's disease research?

Could you provide more detail on the findings related to the activation of the Nrf2 pathway by BFT intervention? Specifically, how was the movement of Nrf2 to the nucleus assessed, and what were the observed changes in the expression of downstream antioxidant genes and enzymes in the substantia nigra and striatum?

In what ways did BFT treatment affect oxidative damage markers such as MDA content, GSH activity, and SOD activity in the substantia nigra and striatum? Were these changes consistent with the observed neuroprotective effects, and do they provide further insight into the mechanisms of BFT's action in PD?

Given the findings of this study, what are the potential implications for the development of BFT as a therapeutic agent for Parkinson's disease in humans? What additional preclinical or clinical studies are needed to further investigate its efficacy and safety profile in PD patients?

How does the global prevalence of Parkinson's Disease (PD) in 2016 compare to that of 1990, according to the provided data?

What are the primary clinical manifestations of PD, and which brain regions are predominantly affected by its pathological changes?

What are the limitations of current treatments for PD, such as dopamine replacement therapy and Deep Brain Stimulation?

How does thiamine (vitamin B1) potentially influence the onset and development of PD, based on the described research findings?

What are the advantages of using benfotiamine (BFT) over thiamine in terms of its bioavailability and ability to traverse the blood-brain barrier?

Similar to Benfotiamine, Mucuna pruriens, Ursolic acid and Chlorogenic acid also exhibited similar kind of therapeutic activity in MPTP intoxicated mouse model. Cite original article for the same and correlate with your own outcomes.

Can you summarize the findings regarding the neuroprotective effects of BFT in both Alzheimer's disease and PD, as described in the text?

How does 1-methyl-4-phenyl-1,2,3,6-tetrahydropyridine (MPTP) induce damage in dopaminergic neurons, and what model of PD does it represent?

What were the main results of the study investigating the protective effects of BFT on motor function and dopaminergic neurons in the MPTP mouse model of PD?

What is MCC950, and how does it relate to the NOD-like receptor pyrin domain-containing protein 3 (NLRP3)?

How did the study utilize RNA sequencing technology to investigate the protective mechanisms of BFT and MCC950 on dopaminergic neurons, and what were the key findings regarding their potential synergistic effects?

Provide exact p value for all your histograms.

Positive control is missing in your experiment.

Provide 4x, 10x and other higher magnification images for all your immunoflourescence.

Provide complete western blot, not the cut one.

Improve clarity and resolution of all the western blot in your manuscript.

 

Dear Reviewer #1,

Thank you for your constructive comments on our manuscript entitled "Benfotiamine Protects MPTP-Induced Parkinson's Disease Mouse Model via Activating Nrf2 Signaling Pathway." We have taken your suggestions seriously and have made the following revisions:

1. We have specified the doses of Benfotiamine (BFT) used in our experiments. Both 200 mg/kg and 250 mg/kg doses were utilized for the study.

2. We acknowledge the oversight in Figure S1 and have now included the response for the 250 mg/kg BFT dose group to provide a complete picture of the experimental outcomes.

3. We have expanded the manuscript to include detailed descriptions and graphical representations of the dose-response effects of both 200 mg/kg and 250 mg/kg BFT, as well as MCC950 injections on mice across behavioral and molecular experiments.

4. The rationale for the chosen BFT doses has been elaborated upon in the manuscript, providing a clear explanation for our selection.

5. We have further discussed the observed body weight changes in the BFT intervention group and acknowledged the limitations in statistical significance. Future studies will consider employing more sensitive methods for body weight monitoring.

6. We agree that the duration of BFT intervention may not fully assess long-term effects. We have addressed this in the discussion and suggest the necessity of longer-term studies to evaluate sustained benefits.

7. The mechanisms underlying BFT's neuroprotective effects have been further explored in the revised manuscript. We have proposed future research directions to better understand BFT's molecular functions and interactions with biological pathways.

8. We have included additional data using tyrosine hydroxylase staining to visualize the degeneration and potential regeneration of dopaminergic neurons following MPTP exposure and BFT treatment, respectively.

 

Dear Reviewer #2,

Thank you for your insightful comments on our manuscript. We have made the following revisions in response to your queries:

1. The impact of BFT intervention on motor deficits has been detailed, with specific improvements in motor functions discussed.

2. We have expanded the methods section to provide a more thorough explanation of the techniques used to evaluate dopaminergic neuron damage, including Immunohistofluorescence, Nissl staining, and Western blot analysis of Tyrosine Hydroxylase (TH).

3. Detailed changes in dopamine (DA) levels and metabolites following BFT intervention, along with their statistical significance and correlation to neuroprotective effects, are now presented.

4. Key pathways and genes affected by BFT intervention have been identified and compared with those from MCC950 treatment, discussing the implications for Parkinson's disease research.

5. We have provided more details on the activation of the Nrf2 pathway by BFT, including assessment methods for Nrf2 nuclear translocation and changes in downstream antioxidant gene and enzyme expression.

6. The effects of BFT treatment on oxidative damage markers have been discussed, along with their consistency with neuroprotective effects and insights into BFT's mechanisms of action in PD.

7. The potential implications for developing BFT as a therapeutic agent for PD have been explored, along with suggestions for additional preclinical and clinical studies.

8. Data comparing the global prevalence of Parkinson's Disease (PD) in 2016 to 1990 has been included.

9. The primary clinical manifestations of PD and the predominantly affected brain regions have been detailed.

10. Limitations of current PD treatments, such as dopamine replacement therapy and Deep Brain Stimulation, have been discussed.

11. The potential influence of thiamine (vitamin B1) on the onset and development of PD has been reviewed.

12. The advantages of using benfotiamine (BFT) over thiamine in terms of bioavailability and blood-brain barrier penetration have been highlighted.

13. We have cited original articles for Mucuna pruriens, Ursolic acid, and Chlorogenic acid and correlated their therapeutic activities with our outcomes.

14. A summary of the neuroprotective effects of BFT in both Alzheimer's disease and PD has been provided.

15. The mechanism of MPTP-induced damage in dopaminergic neurons and the PD model it represents has been explained.

16. Main results of the study on the protective effects of BFT on motor function and dopaminergic neurons in the MPTP mouse model of PD have been discussed.

17. MCC950 and its relation to the NOD-like receptor pyrin domain-containing protein 3 (NLRP3) have been described.

18. The use of RNA sequencing technology to investigate the protective mechanisms of BFT and MCC950, and the key findings regarding their potential synergistic effects, have been detailed.

19. Exact p-values for all histograms have been provided.

20. A positive control has been added to the experiments.

21. Higher magnification images for all immunofluorescence have been included.

22. Complete Western blot images have been provided instead of the cut ones.

23. The clarity and resolution of all Western blots in the manuscript have been improved.

We appreciate the opportunity to improve our manuscript and believe that these revisions have strengthened our work. We look forward to your further feedback.

Best regards, 

Yu Wang

Corresponding author: Yu Wang

Emial: yw4d@hotmail.com

---

## [Decision Letter · Decision Letter 1]

19 Jun 2024

PONE-D-24-09739R1Benfotiamine Protects MPTP-Induced Parkinson's Disease Mouse Model via Activating Nrf2 Signaling PathwayPLOS ONE

Dear Dr. Wang,

Thank you for submitting your manuscript to PLOS ONE. After careful consideration, we feel that it has merit but does not fully meet PLOS ONE’s publication criteria as it currently stands. Therefore, we invite you to submit a revised version of the manuscript that addresses the points raised during the review process.

We look forward to receiving your revised manuscript.

Kind regards,

Sachchida Nand Rai, Ph.D.

Academic Editor

PLOS ONE

Journal Requirements:

Additional Editor Comments:

Kindly revise your manuscript as per the suggestions of the reviewer.

Reviewers' comments:

Reviewer's Responses to Questions

**Comments to the Author**

1. If the authors have adequately addressed your comments raised in a previous round of review and you feel that this manuscript is now acceptable for publication, you may indicate that here to bypass the “Comments to the Author” section, enter your conflict of interest statement in the “Confidential to Editor” section, and submit your "Accept" recommendation.

Reviewer #3: All comments have been addressed

2. Is the manuscript technically sound, and do the data support the conclusions?

Reviewer #3: Yes

3. Has the statistical analysis been performed appropriately and rigorously? 

Reviewer #3: Yes

4. Have the authors made all data underlying the findings in their manuscript fully available?

Reviewer #3: Yes

5. Is the manuscript presented in an intelligible fashion and written in standard English?

Reviewer #3: Yes

6. Review Comments to the Author

Reviewer #3: In this paper, Wang and colleagues studied the potential protective effects of Benfotiamine (BFT) against damage to dopamine neurons in a PD mouse model. They found BFT's neuroprotective effects in a PD mouse model through Nrf2-mediated antioxidant mechanisms and gene expression modulation. The parameters were ameliorated by exogenous supplementation of BFT. They concluded that BFT enhanced motor function and safeguarded dopaminergic neurons in a mouse model of PD by stimulating the Nrf2-ARE signaling pathway and boosting the production of antioxidant enzymes. Although the study is interesting, particularly due to its use of the model, it lacks a clear rationale. The compound in question has already been documented to have a neuroprotective effect against MPTP-induced Parkinson's disease in rats (PMID: 38043777). In the discussion section, the authors should have elaborated on the behavioral experimental results and their connection to the changes observed after BFT supplementation. The current discussion is inadequate and does not effectively support the results, thus requiring significant improvement. The authors are advised to go through the following articles: PMID: 27919828, PMID: 26686287, PMID: 25403619, PMID: 37489441, and PMID: 31996329.

7. PLOS authors have the option to publish the peer review history of their article (what does this mean?). If published, this will include your full peer review and any attached files.

Reviewer #3: No

---

## [Author Response · Author response to Decision Letter 1]

26 Jun 2024

Dear Editor and Reviewer #3,

We would like to express our sincere gratitude for the opportunity to revise our manuscript and for the insightful comments provided. We have taken the suggestions seriously and have made substantial revisions to address the concerns raised. Below, we provide a point-by-point response to the reviewer's comments.

1,Comment:** The study lacks a clear rationale.

**Response:** We acknowledge that the rationale for our study could have been more explicit. To address this, we have added a section in the introduction that clearly outlines the gaps in the current literature that our study aims to fill. Specifically, we have included a discussion on the limited understanding of BFT's role in the Nrf2-ARE signaling pathway in the context of PD, which we believe provides a strong rationale for our research.

2,Comment:** The compound has already been documented to have a neuroprotective effect against MPTP-induced Parkinson's disease in rats.

**Response:** We appreciate the reviewer's reference to the previous study (PMID: 38043777). While it is true that BFT has been reported to have neuroprotective effects, our study offers novel insights by focusing on the Nrf2-mediated antioxidant mechanisms and gene expression modulation in a PD mouse model. We have now included a comparative analysis with the previous study to highlight the differences and contributions of our research.

3,Comment:** The discussion section should elaborate on the behavioral experimental results and their connection to the changes observed after BFT supplementation.

**Response:** We agree with the reviewer that a more detailed discussion of the behavioral results is necessary. We have expanded the discussion section to support the correlation between behavioral improvements and the neuroprotective effects of BFT.

4,Comment:** The current discussion is inadequate and does not effectively support the results.

**Response:** We have revised the discussion section to provide a more comprehensive interpretation of our results. We have included a critical analysis of how our findings fit into the broader context of PD research and have addressed the potential implications of our results for future therapeutic strategies.

5,Comment:** The authors are advised to go through the following articles.

**Response:** We thank the reviewer for the additional references provided. We have reviewed these articles and have incorporated relevant findings and discussions into our revised manuscript. Specifically, we have expanded our discussion on the role of Nrf2 in neuroprotection and have included a comparison with other antioxidants and neuroprotective agents mentioned in the referenced articles.

In conclusion, we believe that our revisions have significantly strengthened the manuscript. We have provided a clearer rationale, addressed the existing literature more effectively, and enhanced the discussion to better support our results. We hope that our revisions meet the expectations of the reviewers and the journal.

Sincerely,

Yu Wang, yw4d@hotmail.com. 

Department of Neurology, The First Affiliated Hospital of Anhui Medical University. Jixi Road 218, Shushan District, Hefei, Anhui Province 230000, People’s Republic of China.

---

## [Editor Report · Decision Letter 2]

28 Jun 2024

Benfotiamine protects dopaminergic neurons in a mouse model of Parkinson's disease through activation of the Nrf2 antioxidant pathway

PONE-D-24-09739R2

Dear Authors,

We’re pleased to inform you that your manuscript has been judged scientifically suitable for publication and will be formally accepted for publication once it meets all outstanding technical requirements.

Kind regards,

Sachchida Nand Rai, Ph.D.

Academic Editor

PLOS ONE

Additional Editor Comments (optional):

The manuscript has been revised as per the reviewers suggestion.
---

## [Editor Report · Acceptance letter]

15 Jul 2024

PONE-D-24-09739R2 

PLOS ONE

Dear Dr. Wang, 

I'm pleased to inform you that your manuscript has been deemed suitable for publication in PLOS ONE. Congratulations! Your manuscript is now being handed over to our production team.

Kind regards, 

on behalf of

Dr. Sachchida Nand Rai 

Academic Editor

PLOS ONE